# Determinants of neonatal near misses among neonates admitted to Guji and Borena zones selected public hospitals, Southern Ethiopia, 2021: A facility based unmatched case control study design

**Anteneh Fikrie**[1]*, **Elias Amaje**[1], **Amana Jilo Bonkiye**[2], **Wako Golicha Wako**[1], **Alqeer Aliyo**[3], **Takala Utura**[1], **Nurye Sirage**[2], **Boko Loka**[4]

**1** School of Public Health, Institute of Health, Bule Hora University, Bule Hora, Ethiopia, **2** Department of Midwifery, Institute of Health, Bule Hora University, Bule Hora, Ethiopia, **3** Department of Medical Laboratory Science, Institute of Health, Bule Hora University, Bule Hora, Ethiopia, **4** Department of Nursing, Institute of Health, Bule Hora University, Bule Hora, Ethiopia

* antenehfikrie3@gmail.com

## Abstract

There is little available evidence that quantifies the determinats of NNM in Ethiopia despite an increasing magnitude of neonatal mortality. Therefore, this study was designed to provide concrte evidence about the determinats of NNMS among neonates admitted to Guji and Borena Zones Public Hospitals, Southern Ethiopia, 2021. A facility based unmatched case control study design was conducted on 402 (134 cases and 268 controls) selected neonates admitted to Bule Hora, Adola and Yabelo General Hospitals from February 1-March 31, 2021. Cases were consecutively selected. Whereas for each case, two controls were selected by systematic random sampling technique. The data collection included a pretested and structured face-to-face interviewer administered questionnaire with a supplementation of maternal and neonatal medical records with checklists. Then the data were coded and entered in to Epi data version 3.1 and then exported to the Statistical Package for Social Science IBM version 25 for analysis. The descriptive statistics run and the results of the data were presented using frequencies, and tables. Bivariable and multi variable logistic regression was used for the analsysis of the data. Finally, Adjusted Odds Ratio together with 95% Confidence Intervals and p value <0.05 was used to declare the significance of all statistic. A total of 134 cases (neonatal near misses) and 268 controls (normal neonate) were participated in this study to make a response rate of 100% for both cases, and controls. In this study rural residence (AOR = 0.51, 95% CI: 0.27, 0.96), previous history of neonatal death (AOR = 4.85, 95%CI: 2.24,10.49), birth interval $\leq$ 2 years (AOR = 1.83, 95% CI: 1.04, 3.11) and history of abortion (both induced and miscarriage) (AOR = 1.97, 95%CI: 1.17, 3.31) were found to be statistically significant at a p-value of <0.05. History of prior abortion history of prior neonatal death and short birth interval ($\leq$ 2 years) were identified to be the determinats of NNMs. High quality antenatal and intrapartum continuum of care should be

**Data Availability Statement:** Data essential for the conclusion are included in this manuscript.

**Funding:** Bule Hora University has funded the research (award number: BHU/RPD/262/13). AF has received the award. The funder had no role in study design, data collection and analysis, decision to publish, or preparation of the manuscript.

**Competing interests:** The authors declare that they have no competing interests.

**Abbreviations:** ANC, Ante Natal Care; AOR, Adjusted Odds Ratio; CI, Confidence Interval; COR, Crude Odds Ratio; CS, Cesarean Section; EDHS, Ethiopian Demographic and Health Survey; HIV, Human Immuno deficiency virus; IQR, Inter Quartile Range; LB, Live Birth; LBW, Low Birth Weight; NICU, Neonatal Intensive Care Unit; NMR, Neonatal Mortality Rate; NNM, Neonatal Near Miss; SDGs, Sustainable Development Goals; WHO, World Health Organization.

provided for women and neonates. Additionally, contraceptive utilization should be encouraged for a women to space the births of their children.

## Background

Until now, there is no standard definition for identification of the criteria for neonatal near-miss(NNM) case. The development of criteria to identify NNM cases is challenged by the absence of a gold standard definition for near-miss cases [1]. However, the 2009 published definition of Maternal Near Miss (MNM) enhanced the understanding for development process of the concept and criteria of NNM [2]. The development of criteria and the definition of NNM are important for the subsequent creation of an epidemiological surveillance system to be used as a tool for public policy and case management [2]. Neonatal near miss case is defined as survival to the 7th day of life and presenting a risk condition at birth (5th minute Apgar <7, birth weight <1,750g, or gestational age <33 weeks) [3]. Identification of NNM case is based two group pragmatic and management criteria [1, 2]. The first group is the pragmatic criteria included by the largest WHO study area (Apgar <7, birthweight < 1750 g and gestational age <33 weeks) [4]. The second group was characterized by the following management criteria: Respiratory distress, blood transfusion, presence of infection with clinical concern, Bile stained vomiting, feeding problems severe enough to cause clinical concern, Cardiopulmonary resuscitation, Congenital Malformations, Convulsion, Surgery, Phototherapy within 24 hours of life, Parenteral intravenous drugs or nutrition, Any intubation [2].

Globally in 2017, 2.5 million babies died from preventable causes like prematurity, complication during the time of birth, bacterial infections, congenital malformations, and poor quality or no health care given at all. Almost all neonatal deaths (98%) occur in low- and middle-income countries, with 78% in Southern Asia and sub-Saharan Africa [5]. Every year, 30 million new born, require specialized or intensive care in a hospital; those who survive often do so with preventable conditions and disabilities that will affect them for life [6]. In Ethiopia, the magnitude of NNM case was found to be 23.3% (233 per 1000 live births) at Northern Ethiopia [7] and 6.2%- 33.4% at South Ethiopia [8, 9].

The Sustainable Development goal (SDG) target to end preventable neonatal deaths obliges all countries to reduce the neonatal mortality rate to 12 deaths or less per 1000 live births by 2030 [10]. Similary, Ethiopia has envisioned ending all preventable newborn and child deaths by 2035 [11] and also the country is trying to increase access to effective coverage of life-saving, high impact neonatal and child health interventions through national-level plans such as Growth and Transformation Plan (GTP) and Health Sector Transformation Plan (HSTP) [12]. Moreover, the National Newborn and Child Survival Strategy (2015/16-2019/20), which was part of the HSTP aimed to reduce NNM from 28/1,000 live births in 2015 to 11/1,000 lives birth by the end of 2020 [11]. However, despite 67% reduction in under-five mortality [13], the NMR has remained 29/1000 lives births by the end of 2019 [14]. Evidence showed that adherence to essential newborn care would benefit newborns, adding special and intensive care services would reduce neonatal mortality by 50% [15]. Thus, identification and correction of factors that may improve maternal and neonatal care are more likely to contribute to the reduction in neonatal mortality rate [4].

Despite the burden of the problem only few researches were conducted on neonatal near miss, even those cross sectional studies that were conducted so far were unable to indentify the determinats NNM cases. Moreover, in the study setting there is no available data that

quantifies detrrminats of NNM cases and also in the study area neonatal morbidity and mortality is still high. So that, this study helps us to identify the determinats of NNM which in turn contributed evidence for a better solutions and give possible recommendation for health planners to achieve Ethiopia's 2035 goal to minimize neonatal death.

## Methods

### Study setting and period

The study was conducted in Bule Hora, Yabelo and Adola General Public Hospitals which were found in West Guji, Borena and East Guji Zones respectively from February 1-March 31, 2021. Bule Hora General Hospital is located in Bule Hora town which is the capital of West Guji Zone of the Oromia Region and located at 467 km to the South of Addis Ababa. Bule Hora General Hospital provides health services for 1,389,821 population. According to the 2019/20 Zonal health department Health Management Information System report, the Hospital has an annual delivery of 3250. It has also 186 health professionals. Likewise, Adola General hospital is found at Adola, a town administration equivalent to woreda of East Guji Zone, is located at 470 km to the South of Addis Ababa. It provides health services for 771,879 populations and it has 132 health professionals. The 2019/20 annual delivery report of Adola General Hospital was 2362. Yabelo General Hospital is found at Yabelo twon, a capital city of Borena Zone, is located at 570 km to the South of Addis Ababa. also provides health services for 926,690 populations of Borena Zone and The 2019/20 annual delivery report of Yabelo General Hospital was 3206. All of three hospitals have functional neonatal intensive care units.

### Study design and population

A facility based unmatched case control study was carried out among live births neonates who were admitted to post-natal or neonatal wards within 28 days of birth in Bule Hora, Adola and Yabelo General Hospitals during the study period.

**Selection of cases (neonatal near niss).**   Neonatal near misses were identified by well-trained and experienced data collectors using the standard WHO recommended pragmatic and or management criteria but survived this condition within the first 27 days of life. The pragmatic criteria are: Birth weight less than 1750g, Gestational Age less than 33 weeks and Apgar score less than 7 at 5 minutes. Whereas, management severity criteria are; respiratory distress, blood transfusion, presence of infection with clinical concern, Bile stained vomiting, feeding problems severe enough to cause clinical concern, Cardiopulmonary resuscitation, Congenital Malformations, Convulsion, Surgery, Phototherapy within 24 hours of life, Parenteral intravenous drugs or nutrition, any intubation [4]. Hence, live birth neonates who had at least one components of pragmatic and management criteria was included in the study. Whereas, multiple pregnancies and neonatal deaths, mothers of neonates who gave birth at home and neonates whose mothers died, critically ill and unable to communicated during the study period were excluded.

**Selection of controls (normal neonate).**   Neonates who were admitted to post-natal and neonatal ward in Bule Hora, Yabelo and Adola General hospitals who had no components of pragmatic and management criteria to identify NNM cases confirmed by the senior health care provider (Pediatrician or Neonatologist or Gynecologist or General Practitioner or Health Officer who were working at pediatrics and Obestetrics ward were included in the study as a control. Multiple pregnancies, mothers of neonates who gave birth at home and neonates whose mothers died, critically ill and unable to communicated during the study period were excluded.

## Sample size determination and sampling procedure

Sample size was calculated using Epinfo software version 7.1.4.0 by considering the proportion of controls exposed 15.8%, proportion of cases with exposure 5.4%, two-sided confidence level 95%, power 80% and ratio of controls over cases 2:1 [16]. Then by substituting the above figures in to the Stat Calc, 122 cases and 244 controls were obtained. Then after adding of 10% for the potential none response rate the final minimum calculated sample size became 402 (134 cases and 268 controls). Three governmental hospitals namely; Bule Hora General Hospital, Yabelo General Hospital and Adola General Hospitals were selected from the three Zones based on the availability of NICU service. Then the calculated sample size were allocated for each hospitals proportionally based on the last year annual delivery report. As the number of NNM case was rare, all the identified cases were taken until the sample size gots its target during the study period. On the other hand, for each NNM case, two controls at the same day were selected by systematic random sampling.

## Data collection tools techniques and quality assurance

The data collection tools were adapted from different previous peer reviewed studies [8, 16–18]. The first version of questionnaire was prepared in English language (S1 Text) and then translated to Affaan Oromoo language which is the working language of the Oromiya region. Then it was retranslated to English language to check the consistencies of the information. Data were collected by a pretested and structured face-to-face interviewer administered questionnaire was applied to the mothers and also supplemented with visual observation of medical records of mothers and neonates. The data collection tools consist of socio-demographic and economic characteristics of mothers, obstetrics and medical history of mothers and neonatal characteristics. Six BSc holder Nurses and Midwives were deployed as the data collectors and three MPH holder professionals were recruited as a supervisors. Following a two days training, data collectors and supervisors were commenced the data collection process. Obstetrics and medical history of mothers and neonatal characteristics like pragmatic and managmenet craterias were extracted from medical records of the mothers and neonates respectively by standard checklist. Every data collection day, data collectors and supervisors were met and checked the completeness and consistency of the collected data. The completeness and consistency of the data were checked by investigators.

## Data processing and analysis

First the completeness and consistencies of the data were checked. The data template format were prepared and entered into Epidata version 3.1 and then exported to SPSS version 25 for further analysis. Descriptive statics were run using percentages for categorical data and median with IQR for continuous variables. The data were presented by statistical tables. Covariates included were: maternal age [15–24, 25–34 and ≥35], marital status [Married and Others* (single, divorced and widowed], place of residence (Urban, rural), maternal occupation (House wife, Gov't employed, merchant and Others# (student, daily laborer)) paternal occupation (Farmer, Gov't employed, merchant and others#(student, daily laborer)), maternal education (No education, primary education, secondary education and college and above), paternal education (No education, Primary education, Secondary education and College and above), Number of children (≤2, 3–5 and ≥5), Type of Pregnancy (Unplanned and Planned), Parity (Multiparous and Nulliparous and Primiparous), History of abortion (both induced & miscarriage) (Yes and No), History of neonatal death (Yes and No), Birth interval (Yes and No), ANC (Yes and No), Number of ANC (Yes and No), Gestation at first visit ANC visit (≤12 weeks and >12 weeks), Mother referred from other facility (Yes and No), Premature rupture

of membrane (Yes and No), Mode of delivery (Instrumental deliveries, Spontaneous Vaginal delivery and Cesarean section), Diagnosed with anemia (Yes and No), Hypertension before current pregnancy (Yes and No), Pregnancy induced hypertension(Yes and No), Syphilis (Yes and No), HIV/AIDS (Yes and No), Severe APH (Yes and No), Severe-preeclampsia (Yes and No), Eclampsia (Yes and No) and Uterine rupture (Yes and No).

The bivariable and multivariable logistic regression were used to identify the determinants of neonatal near miss. All covariates with p value $\leq 0.2$ in bivariate analysis were entered in to multivariate logistic regression for adjusting the potential effects of confounding variables. Hosmer and lomeshow test statistic was used to check the assumption of logistic regression. Multicollinearity were also checked and the VIF was $< 10$. Finally, Adjusted odds ratio with 95% confidence interval and p value $<0.05$ were used to declare the significance of all statistic.

## Ethics approval and consent to participate

The study protocol was approved by the Research and Publication Directorate (RPD) of Bule Hora University. Based on the approval, an official letter was written by RPD to each respected Zones: West Guji Zone Health Departement, East Guji Zone Health Departement and Borena Zone Health Departement. Then each Zone Health Departement has wrote support letter for the study hospitals. Furthermore each Medical Director of the Hospital was wrote a letter to Gynecology or Pediatrics ward for their collaboration. At last the data were collected after guaranteed the confidentiality and informed written consent from each study participants. All the study participants were encouraged to participate in the study and at the same time they were also told that they have the right not to participate.

## Results

### Socio-demographic characteristics of respondents

A total of 134 cases and 268 controls were participated in this study to make a response rate of 100% for both cases, and controls. The median (IQR) age of the neonate's mother was for controls and for cases was equally represented 25 (21–30). The majority of the neonate's mother for cases (97.8%) and controls (91.4%) were married. More than three-in-five (61.9%) of neonate's mother for controls and two-in-five (41%) neonet's mother for controls were rural residents. Pertaining to the educational status of neonate's mother (35.9%) for cases and (24.6%) for controls had no education. Similarly, 26.1% of neonates' fathers for cases and 17.9% for controls had no education (Table 1).

### Obstetric history of mothers

The majority of the neonate's mother for cases (90.3%) and for controls (94%) had planned type of pregnancy. The majority of neonate's mother for cases (61.3%) and for controls (57.5%) were multiparous. Regarding abortion (both induced and miscarriage); the majority of neonate's mother for cases (63.5%) and for controls (80.2%) had no history of abortion. On the otherhand, one-fifth neonate's mother for cases (20.9%) and one-in-twenty for controls (5.2%) had history of neonatal death (Table 2).

### Chronic medical history of neonate's mother

Of the neonates' mother, 27.6% for cases and 21.6% for controls had diagnosed with anemia. About 7.5% of neonates' mother for cases and 4.5% for controls had pregnancy induced hypertension. On the other hand, 15.7% of neonates' mother for cases and 16% for controls had syphilis during their current pregnancy (Table 3).

**Table 1. Socio-demographic characteristics of mothers who gave live birth in Guji and Borena Zones selected Public Hospitals, Southern Ethiopia, 2021.**

| Characteristics | Cases (%) | Controls (%) |
|---|---|---|
| **Age in years** | | |
| 15–24 | 56 (41.8) | 124 (46.3) |
| 25–34 | 60 (44.8) | 109 (40.7) |
| ≥35 | 18 (13.4) | 35 (13.0) |
| **Marital status** | | |
| Married | 131 (97.8) | 245 (91.4) |
| Others* | 3 (2.2) | 14 (8.6) |
| **Place of residence** | | |
| Urban | 51 (38.1) | 158 (59.0) |
| Rural | 83 (61.9) | 110 (41.0) |
| **Maternal Occupation** | | |
| House wife | 93 (69.4) | 150 (56.0) |
| Gov't employed | 14 (10.4) | 46 (17.2) |
| Merchant | 14 (10.4) | 34 (12.7) |
| Others# | 13(9.7) | 38 (14.2) |
| **Paternal Occupation** | | |
| Farmer | 58 (43.3) | 78 (29.1) |
| Gov't employed | 26 (19.4) | 73 (27.2) |
| Merchant | 25 (18.7) | 63 (23.5) |
| Others# | 25 (18.7) | 54 (20.1) |
| **Maternal education** | | |
| No education | 48 (35.9) | 66 (24.6) |
| Primary education | 42 (31.3) | 81 (30.2) |
| Secondary education | 26 (19.4) | 71 (26.5) |
| College and above | 18 (13.4) | 50 (18.7) |
| **Paternal education** | | |
| No education | 35 (26.1) | 48 (17.9) |
| Primary education | 39 (29.1) | 72 (26.9) |
| Secondary education | 27 (20.1) | 55 (20.5) |
| College and above | 33 (24.6) | 93 (34.7) |
| **Number of children** | | |
| ≤2 | 89 (66.4) | 197 (73.5) |
| 3–5 | 25 (18.7) | 48 (17.9) |
| ≥5 | 20 (14.9) | 23 (8.6) |

*Single, divorced, widowed

# student, daily laborer.

## Determinats of neonatal near-misses

Bivariable and multivariable logistic regression was done to identify determinants of NNM cases. In the multivariable model after controlling all the potential confounding variables; place of residence, previous history of neonatal death, birth interval and prior history of abortion (both induced and miscarriage) were found to be statistically significant at a p-value of <0.05. Accordingly, neonates whose mothers were urban resident had 49% reduced odds of experiencing NNM as compared to neonates whose mothers were rural resident (AOR = 0.51, 95% CI: 0.27, 0.96). A neonate's mother who had history of abortion were nearly 2 times

**Table 2. Obstetric characteristics of mothers who gave live birth in Guji and Borena Zones selected Public Hospitals, Southern Ethiopia, 2021.**

| Variables | Cases (%) | Controls (%) |
|---|---|---|
| Type of Pregnancy | | |
| Planned | 121 (90.3) | 252 (94.0) |
| Unplanned | 13 (9.7) | 16 (6.0) |
| Parity | | |
| Nulliparous | 12 (8.9) | 20 (7.5) |
| Primiparous | 40 (29.8) | 94(35.0) |
| Multiparous | 82 (61.3) | 154 (57.5) |
| History of abortion (both induced & miscarriage) | | |
| Yes | 49 (36.5) | 53 (19.8) |
| No | 85 (63.5) | 215 (80.2) |
| History of neonatal death | | |
| Yes | 28 (20.9) | 14 (5.2) |
| No | 106 (79.1) | 254 (97.8) |
| Birth interval | | |
| ≤2 years | 108 (80.5) | 188 (70.1) |
| >2 years | 26 (19.5) | 80 (29.9) |
| ANC | | |
| Yes | 97 (72.4) | 220 (82.0) |
| No | 37 (27.6) | 47 (18.0) |
| Number of ANC (n = 318) | | |
| 1–3 | 52 (38.8) | 112 (50.9) |
| ≥4 | 45 (61.2) | 108 (49.1) |
| Gestation at first visit ANC visit | | |
| ≤ 12 weeks | 28 (28.9) | 62 (28.2) |
| >12 weeks | 69 (71.1) | 158 (71.8) |
| Mother referred from other facility | | |
| Yes | 61 (45.5) | 80 (29.8) |
| No | 73 (54.5) | 178 (70.2) |
| Premature rupture of membrane | | |
| Yes | 53 (39.5) | 109 (40.7) |
| No | 81 (60.5) | 159 (59.3) |
| Mode of delivery | | |
| Spontaneous Vaginal delivery | 96 (71.6) | 200 (74.6) |
| Cesarean section | 30 (22.4) | 59 (22.1) |
| Instrumental deliveries | 8 (6.0) | 9 (3.3) |
| Hospital | | |
| Yabelo | 48 (35.8) | 96 (35.8) |
| Adola | 37 (27.6) | 74 (27.6) |
| Bule Hora | 49 (36.6) | 98 (36.6) |

higher likely of being a NNM cases as compared to mothers who hadn't history of abortion (AOR = 1.97, 95%CI: 1.17, 3.31). The odds of NNM were 4.85 times higher among neonate's whose mother had history of neonatal death as compared to their counter parts (AOR = 4.85, 95%CI: 2.24,10.49). Neonates who had a birth interval of ≤ 2 years were 83% times more likely to be NNM cases as compared to neonates who had a birth interval of > 2 years (AOR = 1.83, 95% CI: 1.04, 3.11) (Table 4).

**Table 3. Chronic medical history of neonate's mother who gave live birth in Guji and Borena Zones selected Public Hospitals, Southern Ethiopia, 2021.**

| Variables | Cases (%) | Controls (%) |
|---|---|---|
| Diagnosed with anemia | | |
| Yes | 37 (27.6) | 58 (21.6) |
| No | 97 (72.4) | 210 (78.4) |
| Hypertension before current pregnancy | | |
| Yes | 17 (12.7) | 37 (13.8) |
| No | 117 (87.3) | 231 (86.2) |
| Pregnancy induced hypertension | | |
| Yes | 10 (7.5) | 12 (4.5) |
| No | 124 (92.5) | 256 (95.5) |
| Syphilis | | |
| Yes | 21 (15.7) | 43 (16.0) |
| No | 113 (84.3) | 225 (84.0) |
| HIV/AIDS | | |
| Yes | 31 (23.1) | 57 (21.3) |
| No | 103 (76.9) | 211 (79.7) |
| Severe APH | | |
| Yes | 9 (6.7) | 5 (1.8) |
| No | 125 (93.3) | 263 (98.2) |
| Severe-preeclampsia | | |
| Yes | 9 (6.7) | 9 (3.3) |
| No | 125 (93.3) | 259 (96.7) |
| Eclampsia | | |
| Yes | 3 (2.2) | 8 (3) |
| No | 131 (97.8) | 260 (97) |
| Uterine rupture | | |
| Yes | 2 (1.5) | 4 (1.5) |
| No | 132 (98.5) | 264 (98.5) |

## Discussion

In this study rural residency, history of abortion, history of neonatal death and short birth interval ($\leq 2$ years) were identified to be the determinats of neonatal near misses. Evidence revealed that birth in rural places decreased the odds of survival in the neonatal period [10]. Our study result also found that neonates whose mothers were urban resident had 49% reduced odds of experiencing NNM as compared to neonates whose mothers were rural resident. This finding is in consistent to a study conducted at Addis Ababa, Ethiopia [19]. Another study in Northwest Ethiopia found that neonates in rural areas were more likely than those in urban areas to have a severe neonatal morbidity [20]. This is due to the fact that rural residents have inadequate accessibility and utilization health services and also have low information on the danger sign and pregnancy related complications. Moreover, mothers in rural areas have less health seeking bahaviour. This implies the need to strengthen the health delivery at the primary health care level, where the majority of womens are served.

In this study a neonate's whose mother had history of abortion (both induced and miscarriage) were nearly 2 times higher likely of being a NNM cases as compared to neonate's whose mothers hadn't had history of abortion. Our result is consistent with sveral previous studies that showed significantly positive association in the risk of NNM among women with a history

**Table 4. Multivariable logistic regression analysis on determinants of neonatal near miss among mothers giving live births at Guji and Borena Zones General Hospitals, Southern, Ethiopia, 2021.**

| Variables | Cases (%) | Controls (%) | COR (95% CI) | AOR (95% CI) |
|---|---|---|---|---|
| Marital status | | | | |
| Married | 131 (97.8) | 245 (91.4) | 2.40 (0.68–8.52) | 4.50 (0.86–23.51) |
| Others@ | 3 (2.2) | 14 (8.6) | 1 | 1 |
| Place of residence | | | | |
| Urban | 51 (38.1) | 158 (59.0) | 0.48 (0.28–0.65) | 0.51 (0.27–0.96)* |
| Rural | 83 (61.9) | 110 (41.0) | 1 | 1 |
| Maternal Occupation | | | | |
| House wife | 93 (69.4) | 150 (56.0) | 1.81 (0.91–3.58) | 1.45 (0.56–3.69) |
| Gov't employed | 14 (10.4) | 46 (17.2) | 0.89 (0.37–2.21) | 0.85 (0.24–2.96) |
| Merchant | 14 (10.4) | 34 (12.7) | 1.20 (0.49–2.91) | 1.26 (0.41–3.79) |
| Others# | 13(9.7) | 38 (14.2) | 1 | 1 |
| Paternal Occupation | | | | |
| Farmer | 58 (43.3) | 78 (29.1) | 1.60 (0.89–2.87) | 0.90 (0.38–2.138) |
| Gov't employed | 26 (19.4) | 73 (27.2) | 0.76 (0.40–1.47) | 0.98 (0.435–2.22) |
| Merchant | 25 (18.7) | 63 (23.5) | 0.85 (0.44–1.66) | 0.96 (0.43–2.16) |
| Others# | 25 (18.7) | 54 (20.1) | 1 | 1 |
| Maternal education | | | | |
| No education | 48 (35.9) | 66 (24.6) | 2.02 (1.05–3.88) | 0.51 (0.14–1.79) |
| Primary education | 42 (31.3) | 81 (30.2) | 1.44 (0.74–2.77) | 0.75 (0.25–2.23) |
| Secondary education | 26 (19.4) | 71 (26.5) | 1.01 (0.50–2.05) | 0.70 (0.25–1.93) |
| College and above | 18 (13.4) | 50 (18.7) | 1 | 1 |
| History of abortion | | | | |
| Yes | 49 (36.5) | 53 (19.8) | 2.33 (1.47–3.71) | 1.97 (1.17–3.31)** |
| No | 85 (63.5) | 215 (80.2) | 1 | 1 |
| History of neonatal death | | | | |
| Yes | 28 (20.9) | 14 (5.2) | 4.79 (2.42–9.46) | 4.85 (2.24–10.49)*** |
| No | 106 (79.1) | 254 (97.8) | 1 | 1 |
| Birth interval | | | | |
| ≤2 years | 108 (80.5) | 188 (70.1) | 1.76 (1.07–2.91) | 1.80 (1.04–3.11)* |
| >2 years | 26 (19.5) | 80 (29.9) | 1 | 1 |
| ANC | | | | |
| Yes | 97 (72.4) | 220 (82.0) | 0.57 (0.35–0.93) | 0.65 (0.35–1.18) |
| No | 37 (27.6) | 47 (18.0) | 1 | 1 |
| Mother referred from other facility | | | | |
| Yes | 61 (45.5) | 80 (29.8) | 1.94 (1.27–3.01) | 1.21 (0.71–2.07) |
| No | 73 (54.5) | 178 (70.2) | 1 | 1 |
| Type of Pregnancy | | | | |
| Planned | 121 (90.3) | 252 (94.0) | 0.59 (0.27–1.26) | 0.60 (0.21–1.70) |
| Unplanned | 13 (9.7) | 16 (6.0) | 1 | 1 |
| Severe APH | | | | |
| Yes | 9 (6.7) | 5 (1.8) | 3.78 (1.24–11.53) | 1.86 (0.51–6.75) |
| No | 125 (93.3) | 263 (98.2) | 1 | 1 |
| Severe-preeclampsia | | | | |
| Yes | 9 (6.7) | 9 (3.3) | 2.07 (0.80–5.34) | 1.76 (0.63–4.93) |

(*Continued*)

**Table 4.** (Continued)

| Variables | Cases (%) | Controls (%) | COR (95% CI) | AOR (95% CI) |
|---|---|---|---|---|
| No | 125 (93.3) | 259 (96.7) | 1 | 1 |

*signifiant at P-value of <0.05

**signifiant at P-value of <0.01

***signifiant at P-value of <0.001

of previous both misccariages and induced abortion [21–24]. This similarity might be due to the fact that women who have history of abortion has suffered from the short and long term consequences, such as cervical insufficiency and uterine adhesions [25]. Studies showed that spontaneous miscarriage has greatly increases the risk of low Apgar score at 1 minute, low birth weight and intrauterine growth [26] and this might be associated with genetic, immunological, infectious or uterine abnormalities [25]. Moreover, studies found that as the number of abortions increases the magnitude of the estimate also increased [24, 25, 27, 28]. This might reflect the need of improving the quality of obstetrics cares offered during pregnancy and delivery, should contribute to reducing deaths of newborns through assuring the quality of care they receive during childbirth.

Prior studies found that neonate's whose mother had history of stillbirth were at higher risk of expriencing an adverse perinatal outcomes in succeeding pregnancies [19, 22, 29, 30]. Our study found a result in line with the above evidence. The odds of NNM were 4.8 times higher among neonate's mother who had history of neonatal death as compared to their counter parts. Similarly, a study conducted at India found that previous history of neonatal stillbirth was associated with risk of neonatal near misses and death [31]. This congruency might be due to the fact that a women who has histry of still birth might be challenged with various psychological symptoms, like stress and depression which persists long after the death of their neonate and and undermines their confidence in achieving future pregnancy success [27]. Moreover, women whose babies have been stillborn feel stigmatised, socially isolated, and less valued by society and the affect could persist for the consequents pregnancies [32]. This implies, the need to emphasize and strengthen high quality antenatal and intrapartum care for women and newly born infants. More importantly, current evidence showed that stillbirth should be used as an indicator of quality of care in pregnancy and childbirth [32].

Neonates who had a birth interval of $\leq$ 2 years were 83% times more likely to be NNM cases as compared to neonates who had a birth interval of > 2 years. This study is corrobotated by studies conducted in Ethiopia [8, 19]. Likewise, this finding is supported by previous studies conducted in Bangladesh [22, 33] and in six low and lower-middle income countries [34]. Moreover, a meta-analysis of 16 studies reported a similar finding [29]. Another similar studies conducted at India [31], UK [35] and China [21] also found the same result. This similarity might be due to the fact that short birth interval has attributed to an increased risk of adverse maternal and perinatal health outcomes [36].

This hospital based unmatched case control study has contributed to the existing knowledge regarding the determinants of NNM cases which can be used by health program planners, policy makers and public health practitioners who are working at improvement of neonatal health and health service at alls. This study conducted at three zonal level of hospitals whixh can enhances. However, the findings from this study would be difficult to infer to the the target population, for the reason that the study was not community based. Furthermore the data were only collected retrospectively, there might be an introduction of recall bias.

## Conclusions

In this study rural residency, history of abortion, history of neonatal death and short birth interval ($\leq$ 2 years) were identified to be the determinats of neonatal near misses. Contraceptive utilization should be encouraged for a women to space the births of their children [11]. Strong public health policies should be established or needs to be reinforced for the provision of high quality essential health care for mothers, and newborns at the community and health facility levels. The local, regional and national governments should strengthened intersectoral collaboration to ensure community empowerment and demand creation for effective use of newborn and child survival interventions with due focus on mothers who had had prior history of adverse perintal outcomes and on the marginalized portions and rural community requiring equitable distribution of services. Further prospective longitudinal study should be conducted by incorporating some missed variables from this study.

## Supporting information

**S1 Text. English version questionnaire.**
(DOCX)

## Acknowledgments

We would like to acknowledge Bule Hora University Research and Publication Directorate Office for allowing us to conduct this usefull study. We are also keen to extend our earnest garitude to the administrators of Bule Hora, Adola and Yabelo General Hospitals for the facilitation during the study period. Our thanks also goes to data collectors, supervisors and those who were actively participated in our study.

## Author Contributions

**Conceptualization:** Anteneh Fikrie, Wako Golicha Wako, Takala Utura.

**Data curation:** Anteneh Fikrie, Wako Golicha Wako, Takala Utura.

**Formal analysis:** Anteneh Fikrie, Wako Golicha Wako, Takala Utura.

**Funding acquisition:** Anteneh Fikrie, Elias Amaje, Wako Golicha Wako.

**Investigation:** Anteneh Fikrie, Elias Amaje, Alqeer Aliyo, Takala Utura.

**Methodology:** Anteneh Fikrie, Elias Amaje.

**Project administration:** Anteneh Fikrie, Elias Amaje, Alqeer Aliyo.

**Resources:** Anteneh Fikrie, Elias Amaje, Amana Jilo Bonkiye, Wako Golicha Wako, Alqeer Aliyo, Takala Utura.

**Software:** Anteneh Fikrie, Amana Jilo Bonkiye, Alqeer Aliyo, Takala Utura.

**Supervision:** Anteneh Fikrie, Elias Amaje, Amana Jilo Bonkiye, Wako Golicha Wako, Alqeer Aliyo, Takala Utura, Nurye Sirage, Boko Loka.

**Validation:** Anteneh Fikrie, Elias Amaje, Amana Jilo Bonkiye, Wako Golicha Wako, Alqeer Aliyo, Takala Utura, Nurye Sirage.

**Visualization:** Anteneh Fikrie, Elias Amaje, Amana Jilo Bonkiye, Wako Golicha Wako, Alqeer Aliyo, Takala Utura, Boko Loka.

**Writing – original draft:** Anteneh Fikrie, Nurye Sirage, Boko Loka.

**Writing – review & editing:** Anteneh Fikrie, Elias Amaje, Amana Jilo Bonkiye, Wako Golicha Wako, Alqeer Aliyo, Takala Utura, Nurye Sirage, Boko Loka.

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
