## [Decision Letter · Decision Letter 0]

31 Aug 2021

 PGPH-D-21-00439 Determinants of neonatal near misses among neonates admitted to Guji and Borena zones selected public hospitals, Southern Ethiopia, 2021. A facility based unmatched case control study design PLOS Global Public Health

Dear Dr. Fikrie,

Thank you for submitting your manuscript to PLOS Global Public Health. After careful consideration, we feel that it has merit but does not fully meet PLOS Global Public Health’s publication criteria as it currently stands. Therefore, we invite you to submit a revised version of the manuscript that addresses the points raised during the review process.

 Please make sure to address all comments in the attachment along with the queries provided below. 

We look forward to receiving your revised manuscript.

Kind regards,

Jenil Patel, MBBS, MPH, PhD

Academic Editor

Journal Requirements:

Additional Editor Comments (if provided):

Reviewers' comments:

Reviewer's Responses to Questions

**Comments to the Author**

1. Does this manuscript meet PLOS Global Public Health’s publication criteria? Is the manuscript technically sound, and do the data support the conclusions? The manuscript must describe methodologically and ethically rigorous research with conclusions that are appropriately drawn based on the data presented.

Reviewer #1: Yes

Reviewer #2: Yes

2. Has the statistical analysis been performed appropriately and rigorously?

Reviewer #1: Yes

Reviewer #2: Yes

3. Have the authors made all data underlying the findings in their manuscript fully available (please refer to the Data Availability Statement at the start of the manuscript PDF file)?

Reviewer #1: Yes

Reviewer #2: Yes

4. Is the manuscript presented in an intelligible fashion and written in standard English?

Reviewer #1: Yes

Reviewer #2: No

5. Review Comments to the Author

Reviewer #1: This is a very well-conducted case-control study in a region of Ethiopia, whose data were collected in 3 large hospitals that had neonatal intensive care units.

According to the authors, there is no study of this type previously carried out, which is one of the points that justifies such research.

The study relied on detailed data collection carried out by trained experts.

The analysis relied on descriptive and inferential statistics (Odds Ratio) to verify the relationships between some variables and the occurrence of the case.

The authors were objective and clear in describing the method and analysis.

The results are well described and presented in well-prepared tables.

The discussion brings to light the congruences and inconsistencies with other studies mentioned and allows us to conclude that sociodemographic variables, as well as the history of previous abortion, can be determinant for the occurrence of the studied cases.

The limitations of the study are exposed by the authors.

In the conclusions, the authors cite the use of contraceptives to increase the time between one pregnancy and another. But, the question remains whether this would be the only or the best alternative to be scored in the conclusions. I believe that it is worth bringing up a little about how public policies are being implemented by the local government, as well as reflecting on the installed capacity of material and human resources to serve this population (the authors bring data on the coverage of services and the quantity of professionals from the study hospitals in the method).

Reviewer #2: Thank you to the authors for studying this important topic. It is important for us to understand how to address the Neonatal mortality to help us reach the SDGs.

The authors need to do a thorough edit for the document- there are several spelling and typographical errors. Some terms need to be better explained.

In general, the paper would benefit from a better review of global data on this topic. If there is insufficient data on neonatal near-miss, maybe the authors can add some relevant information from maternal near miss data and how this approach has been useful in modifying practices

There needs to be a section on data quality issues and what is lacking. Is the sample size sufficient, etc.? The authors should also add a few sentences on policy relevance and need for further investigation.

There are many places within the document where I had a question or a suggestion. I have included that as a sticky note in the document. I have also changed text that you will see in a different font.

6. PLOS authors have the option to publish the peer review history of their article (what does this mean?). If published, this will include your full peer review and any attached files.

**Do you want your identity to be public for this peer review?** For information about this choice, including consent withdrawal, please see our Privacy Policy.

Reviewer #1: **Yes: **Maria Eugenia Firmino Brunello

Reviewer #2: No

---

## [Decision Letter · Decision Letter 1]

2 Nov 2021

PGPH-D-21-00439R1

Determinants of neonatal near misses among neonates admitted to Guji and Borena zones selected public hospitals, Southern Ethiopia, 2021. A facility based unmatched case control study design

Dear Dr. Fikrie,

Thank you for submitting your manuscript to PLOS Global Public Health. After careful consideration, we feel that it has merit but does not fully meet PLOS Global Public Health’s publication criteria as it currently stands. Therefore, we invite you to submit a revised version of the manuscript that addresses the points raised during the review process.

Please address all the comments from Reviewer 3 for additional considerationAdd a response letter along with highlighted manuscript with corresponding changes as asked by Reviewer 3. Provide specific feedback from your evaluation of the manuscript

We look forward to receiving your revised manuscript.

Kind regards,

Jenil Patel, MBBS, MPH, PhD

Academic Editor

Journal Requirements:

Additional Editor Comments (if provided):

Reviewers' comments:

Reviewer's Responses to Questions

**Comments to the Author**

1. If the authors have adequately addressed your comments raised in a previous round of review and you feel that this manuscript is now acceptable for publication, you may indicate that here to bypass the “Comments to the Author” section, enter your conflict of interest statement in the “Confidential to Editor” section, and submit your "Accept" recommendation.

Reviewer #1: All comments have been addressed

Reviewer #3: (No Response)

2. Does this manuscript meet PLOS Global Public Health’s publication criteria? Is the manuscript technically sound, and do the data support the conclusions? The manuscript must describe methodologically and ethically rigorous research with conclusions that are appropriately drawn based on the data presented.

Reviewer #1: Yes

Reviewer #3: Yes

3. Has the statistical analysis been performed appropriately and rigorously?

Reviewer #1: Yes

Reviewer #3: Yes

4. Have the authors made all data underlying the findings in their manuscript fully available (please refer to the Data Availability Statement at the start of the manuscript PDF file)?

Reviewer #1: Yes

Reviewer #3: Yes

5. Is the manuscript presented in an intelligible fashion and written in standard English?

Reviewer #1: Yes

Reviewer #3: Yes

6. Review Comments to the Author

Reviewer #1: The authors adapted the manuscript according to the first review, in particular, in the conclusions raised about public policies that are being implemented or should be implemented, seeking to reduce abortions and neonatal mortality.

Reviewer #3: Authors have described about neonatal near miss and have tried to identify its determinants among neonates admitted to Guji and Borena Zones Public Hospitals, Southern Ethiopia. Authors have appropriately described what is already known about neonatal near miss and they have tried to identify and emphasize the need to understand the determinants of neonatal near miss. The statistical analysis approach is defined clearly; however, I have suggested some changes below that could overall strengthen this article and provide more details in context to the research question.

• In background section, please add reference to the below sentence to support it: “Globally in 2017, 2.5 million babies died from preventable causes like prematurity, complication during the time of birth, bacterial infections, congenital malformations, and poor quality or no health care given at all.”

• Under Data processing and analysis, please elaborate on all the covariates and describe them for better understanding of the model.

• Why was access to healthcare not evaluated for statistical analysis?

Also, I’ve added comments in the article about several typographical errors which authors need to correct.

7. PLOS authors have the option to publish the peer review history of their article (what does this mean?). If published, this will include your full peer review and any attached files.

**Do you want your identity to be public for this peer review?** For information about this choice, including consent withdrawal, please see our Privacy Policy.

Reviewer #1: **Yes: **Maria Eugenia Firmino Brunello

Reviewer #3: No

---

## [Editor Report · Decision Letter 2]

22 Dec 2021

Determinants of neonatal near misses among neonates admitted to Guji and Borena zones selected public hospitals, Southern Ethiopia, 2021. A facility based unmatched case control study design

PGPH-D-21-00439R2

Dear Dr. Fikrie,

We're pleased to inform you that your manuscript has been judged scientifically suitable for publication and will be formally accepted for publication once it meets all outstanding technical requirements.

Within one week, you'll receive an e-mail detailing the required amendments. When these have been addressed, you'll receive a formal acceptance letter and your manuscript will be scheduled for publication.

An invoice for payment will follow shortly after the formal acceptance. To ensure an efficient process, please log into Editorial Manager at https://www.editorialmanager.com/pgph/ click the 'Update My Information' link at the top of the page, and double check that your user information is up-to-date. If you have any billing related questions, please contact our Author Billing department directly at authorbilling@plos.org.

Kind regards,

Jenil Patel, MBBS, MPH, PhD

Academic Editor